# Experimental Investigations on Hydrodynamic Responses of a Semi-Submersible Offshore Fish Farm in Waves

**Yunpeng Zhao [1], Changtao Guan [2], Chunwei Bi [1,*], Hangfei Liu [1] and Yong Cui [2]**

[1]   State Key Laboratory of Coastal and Offshore Engineering, Dalian University of Technology, Dalian 116024, China

[2]   Qingdao Key Laboratory for Marine Fish Breeding and Biotechnology, Yellow Sea Fisheries Research Institute, Chinese Academy of Fishery Sciences, Qingdao 266071, China

*   Correspondence: bicw@dlut.edu.cn

**Abstract:** A series of physical model experiments was performed to investigate the hydrodynamic responses of a semi-submersible offshore fish farm in waves. The structural configuration of the fish farm primarily refers to that of the world's first offshore fish farm, Ocean Farm 1, developed by SalMar in Norway. The mooring line tension and motion response of the fish farm were measured at three draughts. The study indicated that the tension on the windward mooring line is greater than that on the leeward mooring line. As the wave height increases, the mooring line tension and motion responses including the heave, surge, and pitch exhibit an upward trend. The windward mooring line tension decreased slightly with increasing draught. The existence of net resulted in approximately 42% reduction in mooring line tension and approximately 51% reduction in surge motion. However, the heave and pitch of the fish farm increased slightly with the existence of net. It was found that the wave parameters, draught, and net have noticeable effect on the hydrodynamic response. Thus, these factors are suggested to be considered in structural designs and optimization to guarantee the ability of the fish farm to resist destruction and ensure safety of workers during intense waves.

**Keywords:** semi-submersible; offshore fish farm; hydrodynamic response; mooring line tension; motion response

## 1. Introduction

Owing to the limitations of ecological environments and space resources in near-shore areas; offshore aquaculture is becoming growing in the aquaculture industry worldwide. High-density polyethylene (HDPE) net cages are increasingly being used in the aquaculture industry. Generally, an HDPE net cage could be moored in a single or array configuration, which is flexible in industrial applications. However, the volume for aquaculture typically ranges from 10,000 m$^3$ to 30,000 m$^3$, not to mention the deformation induced by currents and waves. So far, more than 20 countries and regions around the world have been devoted to aquaculture field. For instance, France and Norway have cooperated to build an aquaculture vessel with 270 m length and the drainage can reach 10$^5$ t. Spanish companies have designed an aquaculture platform, which can resist to waves of 9 m. Norway, as one of the most developed fisheries countries in the world, has already been leading the world in the field of farming equipment design. Recently, a new type of offshore fish farm, Ocean Farm 1 [1] (see Figure 1), which is mainly made up of steel, with a volume of 250,000 m$^3$ was developed in Norway and built in China. This kind of fish farm may represent the first step toward a new era in offshore aquaculture and address key issues related to sustainable growth in the aquaculture industry. As the aquaculture is marching to the open sea, the power supply of the electrical equipment on the fish farm becomes a

weakness of development. The floaters and columns for a fish farm could be ideally used as a wave energy converter platform. The aquaculture platform of sufficient dimensions can support several wind turbines and realize self-power supply of the offshore fish farm. Overall, both the converter platform of wave energy and offshore wind turbine can be combined with this kind of new type of offshore aquaculture facility. Due to its superior performance, the offshore fish farm considered is accepted and widely used all over the world. Hence, fundamental research on the offshore fish farm is very critical, especially in the initial stage.

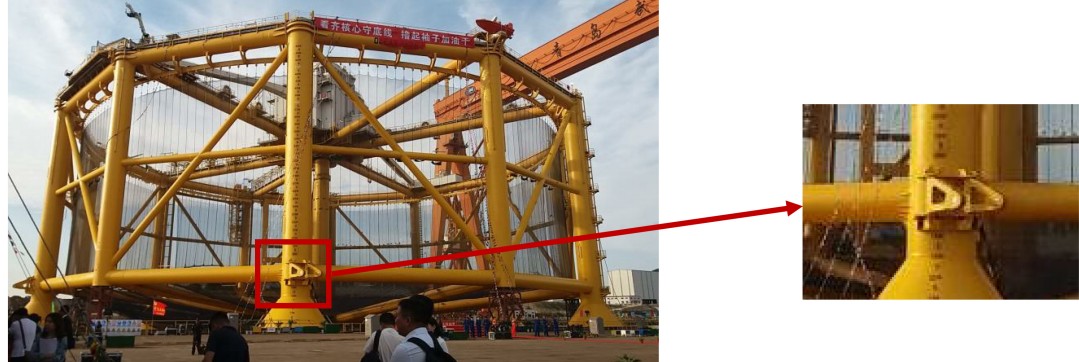

**Figure 1.** Main structure and the location of connection of the ocean Farm 1.

In open seas, a fish farm is easily subjected to strong currents, severe waves, and storm surges, all of which can cause significant destruction. Therefore, studying the hydrodynamic responses of a net cage is important and urgent. In the past few decades, a considerable amount of research including experimental and numerical studies has been carried out to investigate the hydrodynamic responses of offshore net cages. Fredriksson et al. [2] analyzed both the motion response and mooring line tension of a net cage in waves and currents. Lader et al. [3] conducted a series of physical model experiments to study the forces and deformation of a circular net cage with different weights in a uniform flow. Li et al. [4] adopted a numerical model to analyze the influences of a sinker weight on the deformation and uniform load on the net cage; the results of the numerical simulation agreed well with the experimental data. Moe et al. [5] calculated the distribution of loads in the net cage owing to current, weights, and gravity based on commercial explicit finite element software. Tsukrov et al. [6] conducted experimental studies of drag forces on copper alloy net panels and empirical values for normal drag coefficients are proposed for various types of copper netting. Xu et al. [7] applied a statistical approach to determine the motion and tension transfer functions in irregular waves. Fu et al. [8] studied an extended three-dimensional hydro-elasticity theory to predict the dynamic response of $5 \times 2$ floating fish farm collars in waves. Zhao et al. [9] performed a number of physical model experiments to study the main mooring line tensions and flow velocity magnitudes when the current flowed through multiple net cages. Zhou et al. [10] presented hydrodynamic characteristics of knotless nylon netting normal to free stream and the effect of inclination angle on drag coefficient were discussed. Kristiansen et al. [11] investigated the mooring line loads on an aquaculture net cage that was analyzed by model tests and numerical simulations. Huang et al. [12] investigated the dynamic deformation of the floating collar of a net cage under a combination of waves and currents. Yao et al. [13] developed a novel hybrid volume approach to model the current loads on a net cage by considering the fluid–structure interaction. Bi et al. [14] investigated waves propagating through net cages with different levels of biofouling and studied them numerically using a three-dimensional computational fluid dynamics model. In addition, as for the submersible fish cage, Kim et al. [15] used the computational fluid dynamic software to analyze the flow field characteristics of a submersible abalone aquaculture cage and adopted a finite element model based on Morison equation to study the hydrodynamic responses of the moored containment structure. Shainee et al. [16,17] conducted numerical simulations and experimental model tests to examine the submergence characteristics of

a self-submersible SPM cage system in both regular and random waves, respectively. Xu et al. [18] analyzed the hydrodynamic behavior of a self-submersible single-point mooring gravity cage in combined wave–current by numerical simulation.

Although the hydrodynamic responses of the net cage have been studied extensively, studies on new types of offshore fish farms are scarce. Li et al. [19] conducted a preliminary study on the global responses and mooring line loads of a vessel-shaped offshore fish farm in various waves. However, the effect of waves on the hydrodynamic responses of fish farms is an important research issue. Thus, it is essential to analyze the interaction mechanism between fish farms and waves. Unlike traditional net cages, the semi-submersible offshore fish farm is a new type of aquaculture facility that is unique in the sense that draught of the fish farm can be regulated by pontoons. When the waves and winds are severe, the draught of the fish farm can be increased appropriately to avoid destruction; when the fish farm need to be maintained, it can ascend to the sea surface by controlling the water of the pontoon. In addition, owing to the existence of net, the hydrodynamic response is different from the traditional offshore engineering structure. In this study, a series of laboratory experiments were performed in the wave–current flume to study the hydrodynamic characteristics of a semi-submersible offshore fish farm. The physical model of the fish farm, similar to Ocean Farm 1 [1], consists of a primary frame, four mooring lines, nets, pontoons, and sinkers. However, as we do not know the detailed information of Ocean Farm 1, the structural parameters of the fish farm in this study are designed according to our own understanding. The objective of this study is to analyze the hydrodynamic responses of this kind of semi-submersible offshore fish farm and provide positive suggestions for its structural optimization. The present research is based purely on scientific interest and has no commercial purpose.

This paper is organized as follows. The laboratory experiments, including the physical model, experimental setup, experimental conditions, and data analysis methods, are presented in Section 2. Section 3 presents the experimental results of the mooring line tension and motion response in different draughts. A detailed discussion including an analysis on the wave parameters, draught, and net damping is presented in Section 4. Finally, the conclusions are presented in Section 5.

## 2. Laboratory Experiment

In the open sea, a semi-submersible fish farm can be easily subjected to extreme waves. In order to investigate the hydrodynamic responses of a fish farm in regular waves, a series of physical model experiments were conducted. According to the actual sea conditions, prototype structure size, and experimental conditions, the model size, experimental layout, and wave conditions can be designed. All the laboratory experiments were performed in a wave–current flume at the State Key Laboratory of Coastal and Offshore Engineering, Dalian University of Technology, Dalian, China.

### 2.1. Physical Model

The physical model (see Figure 2) consists of the primary frame, net, weight, and mooring systems. The draught of the fish farm can be adjusted by changing the volume of ballast water in the pontoon to determine the working station. In the experiment, it is difficult to completely meet the same similarity law for each part of fish farm, so in order to reduce the influence of the scale effect on the structure, especially the simulation accuracy of the net, the study adopts different similar scales for the net and frame structure. The full scale can be designed by the gravity similarity. The frame and net system adopted two geometric scales, which were 1:120 and 3:40, respectively. More details about the similarity law and the calculation of full-scale values can be found in previous research [20,21].

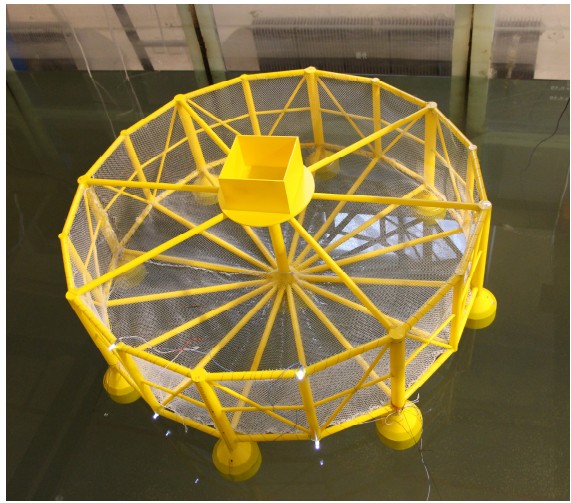

**Figure 2.** Physical model of the semi-submersible offshore fish farm.

### 2.1.1. Frame System

The frame system is the primary structure that bears the environment load and constitutes the fish living space with the net. The frame system of the fish-farm model is made up of plexiglass and includes column, arc pipe, brace, and pontoon (see Figure 3). In accordance with the dimensions of the experimental facilities and the tested wave conditions, the diameter of the fish-farm model is set to 1 m. The detailed parameters of the fish farm in this study are presented in Table 1. The prototype values of the fish farm are also shown with a geometric similarity scale of 1:120, which is roughly similar to the dimensions of Ocean Farm 1 [1]. The deformation of the primary frame of fish farm is small and has a negligible effect on the hydrodynamic responses of the fish farm; therefore, the elastic similarities need not be considered.

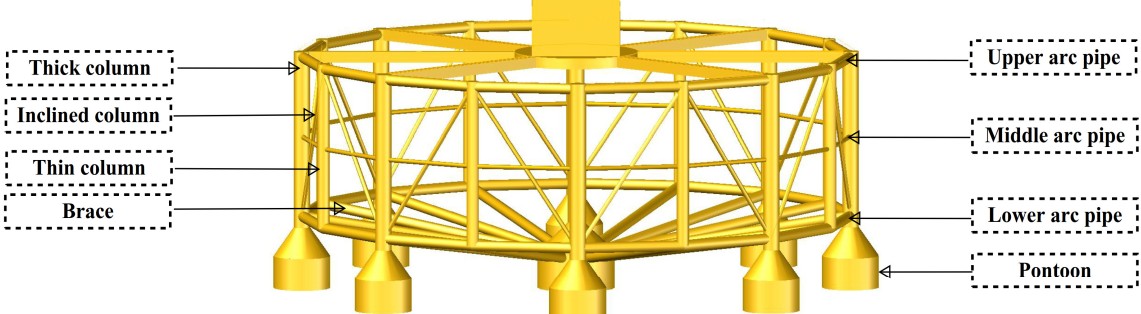

**Figure 3.** Primary frame of the fish farm.

**Table 1.** Structure parameters of the fish farm in this study.

| Component | Parameter | Prototype (m) | Model (m) |
|---|---|---|---|
| Upper arc pipe | Length | 24.00 | 0.20 |
| | Diameter | 1.92 | 0.016 |
| | Thickness | 0.017 | 0.0023 |
| Middle arc pipe | Length | 24.00 | 0.20 |
| | Diameter | 0.96 | 0.008 |
| | Thickness | 0.015 | 0.0021 |
| Lower arc pipe | Length | 24.00 | 0.20 |
| | Diameter | 1.92 | 0.016 |
| | Thickness | 0.017 | 0.0023 |
| Thick column | Length | 36.00 | 0.30 |
| | Diameter | 3.60 | 0.03 |
| | Thickness | 0.025 | 0.0025 |
| Thin column | Length | 33.60 | 0.28 |
| | Diameter | 2.40 | 0.02 |
| | Thickness | 0.030 | 0.0031 |
| Inclined column | Length | 36.00 | 0.30 |
| | Diameter | 0.96 | 0.008 |
| | Thickness | 0.015 | 0.0021 |
| Central column | Length | 36.00 | 0.30 |
| | Diameter | 3.60 | 0.03 |
| | Thickness | 0.025 | 0.0025 |
| Pontoon | Column Length | 7.20 | 0.06 |
| | Height | 6.00 | 0.05 |
| | Diameter | 12.00 | 0.10 |
| | Thickness | 0.040 | 0.005 |
| Brace | Length | 60.00 | 0.50 |
| | Diameter | 1.90 | 0.016 |
| | Thickness | 0.017 | 0.0023 |
| Upper Brace | Length | 60.00 | 0.50 |
| | Diameter | 1.90 | 0.016 |
| | Thickness | 0.017 | 0.0023 |

### 2.1.2. Net System

The net system was designed with three parameters: The overall size, mesh size, and net diameter. To avoid an extremely small net diameter with the model scale of 1:120 such that the model net can hardly be attained, the experimental model primarily considered the drag force to guarantee that the force on the theoretical net is equal to that on the model net (see Figure 4). In the traditional net cage, the whole net system is flexible structure and exhibits a large deformation in regular waves. However, the effect on the deformation of net combined with the fish farm is slight. In the experiment, the net system is designed based on our previous research [20]. The geometric similarity of the net is as follows:

$$\frac{a_1}{d_1} = \frac{a_2}{d_2} \tag{1}$$

where $a_1$ and $d_1$ are prototype mesh size and prototype net diameter, respectively; $a_2$ and $d_2$ are the mesh size and net diameter of the model, respectively.

The prototype mesh size and net diameter are 5 cm and 3.75 mm, respectively. Therefore, it can be calculated that the ratio of the mesh size to the net diameter should be 40/3, such that the drag force on the theoretical net is the same as that on the equivalent experimental net. Through preliminary calculations, the mesh size is 8 mm and the net diameter is 0.6 mm in the experiment.

To guarantee the gravity similarity, the net system is designed based on a previous research study [20]. The gravity similarity of the net is as follows:

$$\Delta W = (\frac{1}{\lambda'} - \frac{1}{\lambda}) \times (\frac{\pi d_p^2}{4 a_p \mu_1 \mu_2} \times 10^4) \times (\rho_n - \rho) \times q \times S \tag{2}$$

$$\lambda' = d_p / d_m \tag{3}$$

where $\Delta w$ is the corrected weight of the net; $\lambda'$ is the small scale of the net that can be calculated by $d_p/d_m$; $\lambda$ is the geometric scale; $d_p$ is the prototype net diameter; $d_m$ is the model net diameter; $\rho_n$ is the net material density; $\rho$ is the water density; $a_p$ is the prototype mesh size; $q$ is the solidity ratio of the net; $S$ is the hanging ratio area of the model net; $\mu_1$ and $\mu_2$ are 0.707 and 0.707, respectively.

Some differences in weight exist between the theoretical net and the equivalent net. To guarantee the weight similarity, it is calculated that the weight of the equivalent net should be increased by 6.7 g, which can be adjusted by the weight system. Owing to the weak elasticity and stiffness of the polyethylene net, the elasticity similarity of the net will not be considered.

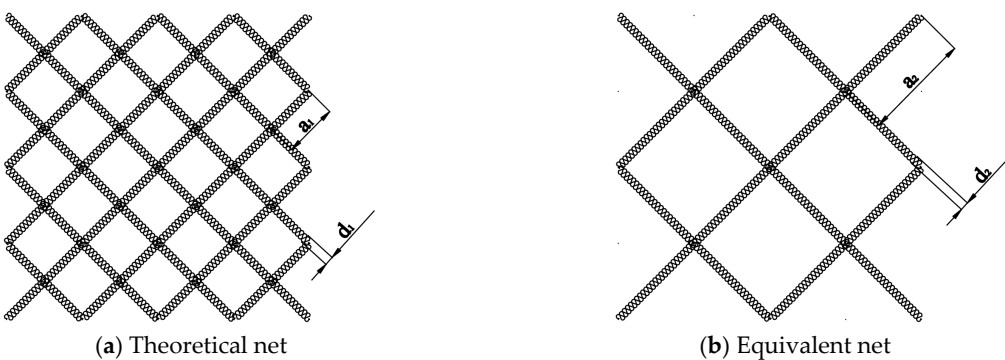

(**a**) Theoretical net  (**b**) Equivalent net

**Figure 4.** Net system similarity.

### 2.1.3. Weight System

To overhaul and operate the fish farm, the draught of the fish farm can be controlled by a weight system (see Figure 5a) to attain a specified state in practical applications. In the experiment, combining the actual state with experimental conditions, the draught can be adjusted by injecting the water into the pontoon (see Figure 5b) and by a sinker (see Figure 5c). In addition, three draughts exist, including those of 7 cm, 28 cm, and 36 cm, representing the maintenance condition and two working conditions of the fish farm, respectively. Meanwhile, the maintenance condition can be reached depending on its own weight; no additional weight is required. However, the working state 1 can be obtained by hanging weights at the bottom of column, injecting water into the pontoon and adding sand to the top box, and the working state 2 needs to continue to add sand into the box to achieving the draught of 36 cm. The detailed conditions can be presented in Table 2.

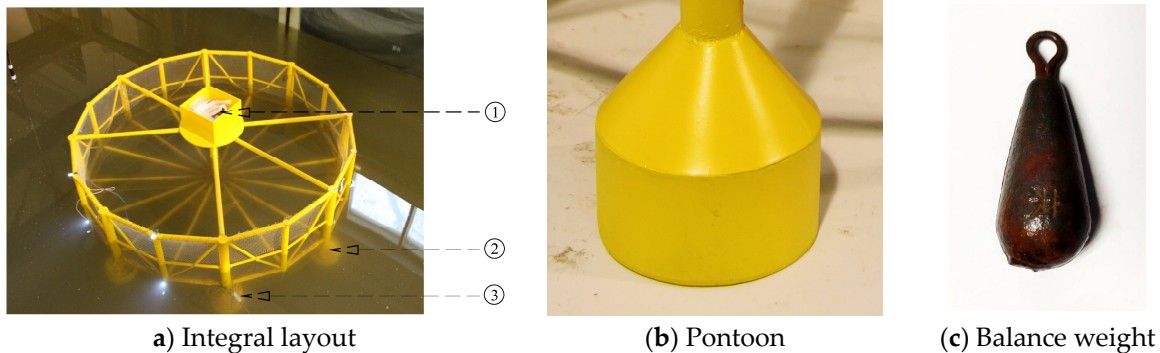

**a**) Integral layout      (**b**) Pontoon      (**c**) Balance weight

**Figure 5.** Weight system. ① Adding sand to the top box; ② injecting water into the pontoon; ③ hanging weights at the bottom of column.

**Table 2.** Weight system of the fish farm.

| Model Conditions | Prototype Draught (m) | Model Draught (cm) | Weights (kg) |
|---|---|---|---|
| Maintenance state | 8.4 | 7.0 | 0 |
| Working state 1 | 33.6 | 28 | 4.02 |
| Working state 2 | 43.2 | 36 | 6.59 |

### 2.1.4. Mooring System

The mooring system is an important part of the fish farm and is crucial for the fish farm to operate steadily in the sea. In the experiment, a four-point anchor form was adopted in the fish farm and the initial tension could be loaded in the mooring line. In the practical application, the wave loads on the mooring line are smaller than those on the fish farm; therefore, the elastic similarity was considered primarily in the experiment such that the model mooring line is similar to that of the prototype and to reduce test errors. The elastic part of the anchor is modeled by the spring, and the anchor rope can be fixed by lead blocks at the bottom. The relationship between the force and the calculated value of elongation of the mooring line can be calculated by Formula (4). The results can be listed in the Table 3.

$$F_m = \frac{C_p d_p^2 (\Delta S/S)^n}{\lambda^3} \tag{4}$$

where $F_m$ is the model cable tension; $C_p$ is the elastic coefficient of prototype cable, which is $26.97 \times 10^4 MPa$; $d_p$ is the diameter of prototype cable, which is 120 mm; $\Delta S/S$ is the elongation rate of prototype cable; $n$ is the coefficient, which is 1.5; $\lambda$ is the geometric scale.

**Table 3.** Force and calculated value of elongation.

| Fm (N) | 1 | 2 | 3 | 4 | 5 | 6 | 7 | 8 |
|---|---|---|---|---|---|---|---|---|
| $\Delta S/S$ | 0.34% | 0.54% | 0.71% | 0.85% | 0.99% | 1.12% | 1.24% | 1.36% |

As listed in Table 3, the relationship between the force and the calculated value of elongation of the mooring line is close to linear. Therefore, the elasticity of the mooring line can be simulated by the spring in the experiment. To obtain a suitable spring, a variety of springs were tested (see Table 4). It can be concluded that the mooring line of the model can be simulated by a polyethylene rope and spring. The mooring line was 3.0 m long in the physical model experiment.

**Table 4.** Elongation rate of spring under different loads.

| Load (N) | Elongation (cm) | Length of Mooring Line (cm) | Elongation Percentage (%) |
|---|---|---|---|
| 1 | 1.2 | 300 | 0.4 |
| 2 | 1.6 | 300 | 0.53 |
| 3 | 2.0 | 300 | 0.67 |
| 4 | 2.4 | 300 | 0.80 |
| 5 | 2.9 | 300 | 0.97 |
| 6 | 3.3 | 300 | 1.10 |
| 7 | 3.7 | 300 | 1.23 |
| 8 | 4.1 | 300 | 1.37 |

The comparison between the calculated elongation of the mooring line and the measured values is shown in Figure 6. It is indicated that the error is small; therefore, the method of simplifying the mooring line is feasible.

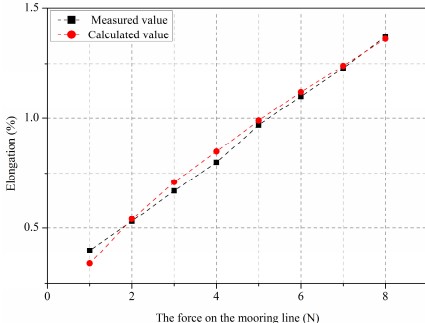

**Figure 6.** Comparison on mooring line elasticity of calculated values and measured value.

## 2.2. Experimental Setup

The wave–current flume is 69 m long, 2 m wide, and 1.8 m deep, and the water depth was 1 m during the experiments. In addition, the flume is equipped with a servo motor-driven, piston-type wave-maker capable of producing regular and irregular waves. At the end of the flume, wave absorbers are installed to mitigate the wave reflection. Both sides of the flume at the working section are smooth glass to reduce viscous dissipation owing to the boundaries. In this flume, both waves and currents can be generated. However, only regular waves were generated in this physical model experiment.

Figure 7 shows a sketch of the experimental setup. The fish farm model was moored by four mooring lines in its equilibrium position. Each mooring line primarily comprises one 2.9 m polyethylene rope whose diameter is 1.0 mm and one 0.1 m stainless spring. The linear density of the rope is 0.003 kg/m and the mass of the spring is 0.01 kg. To measure the forces acting on the mooring lines, four load cells were connected to the windward and leeward mooring lines.

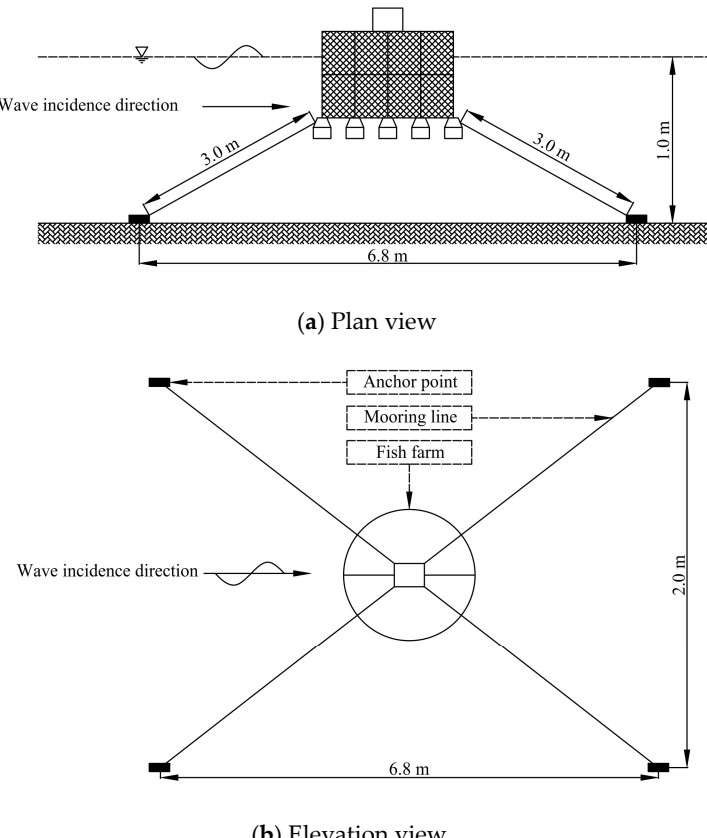

(**a**) Plan view

(**b**) Elevation view

**Figure 7.** Experimental layout.

To obtain the motion response, a charge-coupled device (CCD) camera was used to record the motion trajectory of the diodes that can be fixed on the fish farm, and the motion responses of the fish farm can be calculated by a self-developed software DUT-FlexSim. The motion response can be collected by the CCD high-speed acquisition camera that is arranged in the observation area of the flume (see Figure 8). To guarantee the accuracy of data acquisition of the motion response, the surrounding light should be maintained dark. Four light-emitting diode (LED) bulbs are arranged at the top and bottom of the fish farm as tracing points (see Figure 9). The tracing points along the *x*-axis direction are called the front point, and the tracing points on the back side are called the back point.

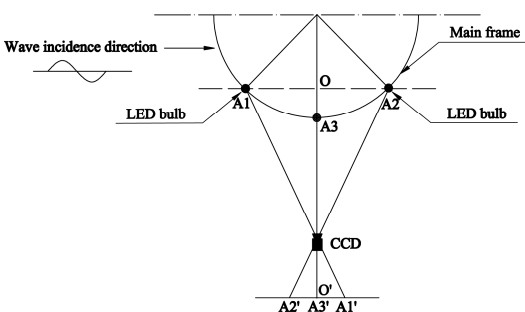

**Figure 8.** Charge-coupled device (CCD) high-speed camera setup.

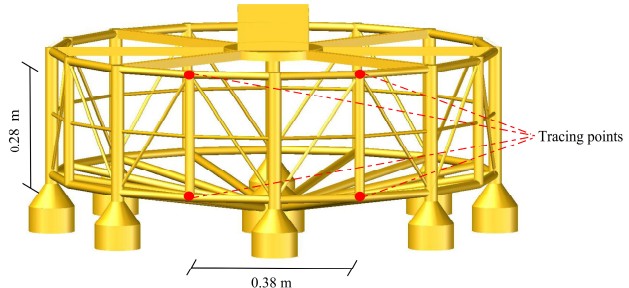

**Figure 9.** Location of tracing points.

## 2.3. Experimental Conditions

In the physical model experiment, the hydrodynamic response of fish farm can be measured in regular waves. According to the model scale, the experimental water depth can be designed to be 1.0 m and its corresponding prototype water depth is 120 m. The mooring line tension and motion response can be measured by the load cells and CCD, respectively. To guarantee the accuracy of data collection, the measurements were performed thrice, and the sampling rate was 0.02 s. Three draughts exist, including those of 7 cm, 28 cm, and 36 cm. In addition, the regular waves were designed as per the specifications in Table 5.

**Table 5.** Wave conditions.

| Wave Case No. | Prototype Value | | Model Value | |
|:---:|:---:|:---:|:---:|:---:|
| | Wave Height (m) | Wave Period (s) | Wave Height (m) | Wave Period (s) |
| 1 | 7.2 | 10.95 | 0.06 | 1.0 |
| 2 | 7.2 | 13.15 | 0.06 | 1.2 |
| 3 | 7.2 | 15.34 | 0.06 | 1.4 |
| 4 | 9.6 | 10.95 | 0.08 | 1.0 |
| 5 | 12 | 10.95 | 0.10 | 1.0 |
| 6 | 12 | 13.15 | 0.10 | 1.2 |
| 7 | 12 | 15.34 | 0.10 | 1.4 |

## 2.4. Data Analysis Method

In the experiment, the water surface elevations were recorded by capacitance-type wave gauges arranged along the center line of the wave flume. The absolute accuracy of these wave gauges is approximately ±1 mm. Before initiating any measurements, the wave gauges were examined for soundness, cleaned if necessary, and subsequently calibrated. In addition, a computer control system developed by the Beijing Hydraulic Research Institute was used for collecting the data of free surface elevations with multiple channels. To collect accurate data, the time series of wave elevation at each measurement point was recorded with a sampling rate of 50 Hz and a stable data over a period of 10 s was chosen for data analysis; subsequently, the wave height at a measurement point was the average value of the corresponding time series.

Water-resistant load cells with a capacity of 10 N were used to measure the forces on the mooring lines, and the specified accuracy of the load cell was 0.1 N. Each measurement was performed thrice to diminish the impact of random and bias errors. Data sampling was conducted over a period of 20 s and the final experimental value was the average value of the three measurements. Figure 10 is the time series of mooring lines tensions when the wave height is 10 cm and the wave period is 1.4 s in a draught of 36 cm. The pretension of mooring line was measured before each group of tests began, and then the total force on the mooring line can be obtained by the load cells. Therefore, the mooring line tension in pure waves can be calculated as follows.

$$F_w = F_t - F_{pre} \tag{5}$$

where $F_w$ is the average value of maximum mooring line tensions in pure waves; $F_t$ is the average value of maximum mooring line tensions which can be measured by water-resistant load cells; $F_{pre}$ is the pre-tension of mooring line. The windward and leeward pre-tension are 1.91 N and 1.87 N, respectively.

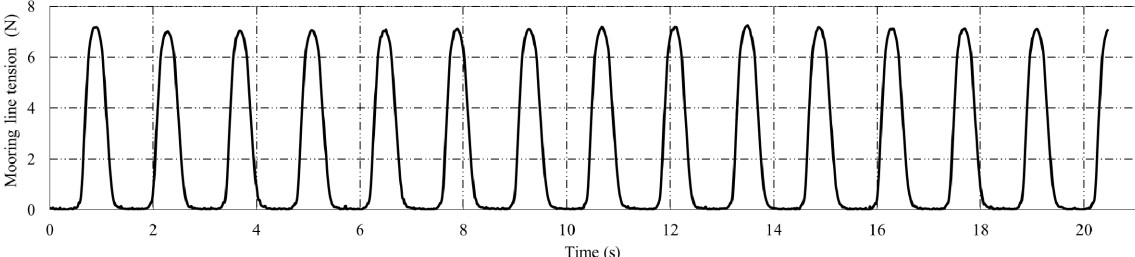

**Figure 10.** The time series of mooring lines tensions.

## 3. Results

In an actual operation of the fish farm, the mooring line tension is the key factor affecting its stability. In addition, the motion response contributes significantly to the safety of the workers. Therefore, the mooring line tension and the motion response can be measured and analyzed in different draughts. The maximum force on the mooring line and the maximum motion response were obtained from time intervals in duration of approximately 15 wave periods from the measured time series.

### 3.1. Mooring Line Tension

Wave period and wave height are two important factors for the mooring line tension. Figures 11 and 12 show the mooring line tension in different wave periods and wave heights, respectively. The primary part of the fish farm is above the sea level, and only nine pontoons of the fish farm are underwater with the draught of 7 cm for maintenance condition, which is easily subjected to the wave load. On the contrary, most of the fish farm is submerged beneath the free surface when the draughts are 28 cm and 36 cm.

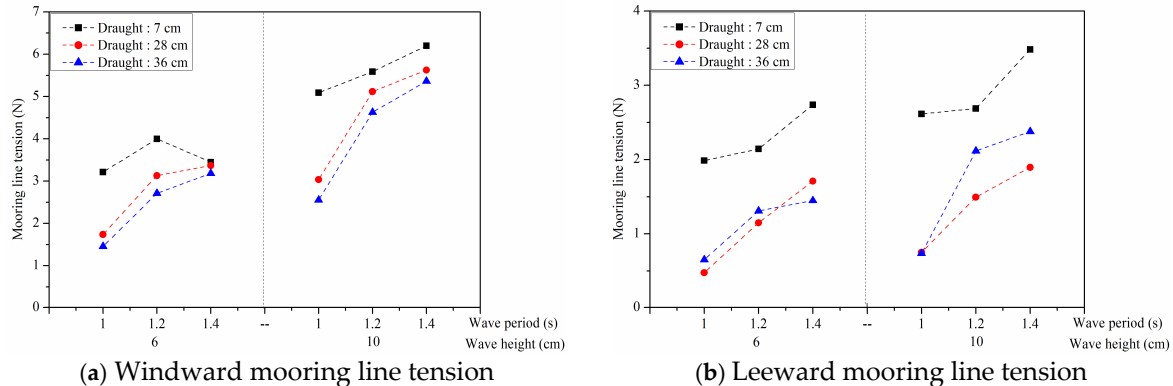

(**a**) Windward mooring line tension      (**b**) Leeward mooring line tension

**Figure 11.** Mooring line tension in different wave periods.

Because of the symmetry of the fish farm model and the mooring lines, the average value of the tension in the mooring lines at symmetrical positions was used for force analysis. The maximum force on the mooring line can be measured by the average value of the two mooring lines in the symmetrical position, and the pretension can be loaded in the mooring line before the experiment begins. It can be observed from Figure 11 that the force on the mooring line is closely proportional to the wave period. Furthermore, as the draught increases, the mooring line tension exhibits a downward trend. It is apparent from Figure 12 that the windward and leeward mooring line tensions exhibit an upward trend with increasing wave height.

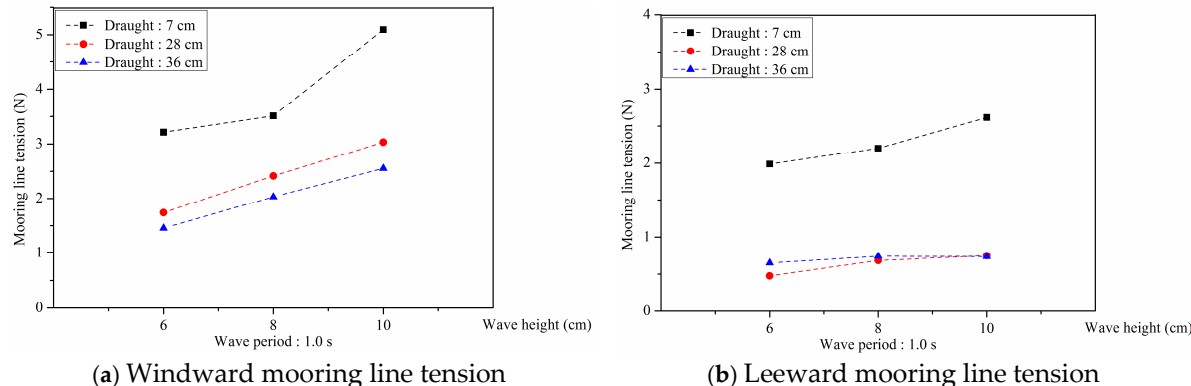

(**a**) Windward mooring line tension　　　　　(**b**) Leeward mooring line tension

**Figure 12.** Mooring line tension in different wave heights.

### 3.2. Motion Responses

In the semi-submersible offshore fish farm, not only the mooring line tension, but also the motion responses are necessary to be analyzed for the fish farm. It is because the motion response is critical to the fish living in the fish farm and human activities on the fish farm. Smaller motion response can provide a stable living space for fish and avoid damage to fish in severe currents and waves, and further guarantee the quality of fish and the economic income of fishermen. The fish farm exhibits a small deformation under waves and is approximately a rigid structure. Therefore, only the motion responses of the fish farm require analysis, which include three aspects: The heave, surge, and pitch. In the present experiment, the fish farm model was fitted with a tracking point on the upstream side and the back side of the fish farm such that the motion trajectory can be analyzed by tracking the two points.

Figure 13 shows trajectory of the fish farm at $H = 10$ cm and $T = 1.4$ s in the draughts of 28 cm and 36 cm. As shown in the Figure 13, the trajectory of the windward side is oblique and elliptical, while the trajectory of the leeward side resembles an egg shape. Meanwhile, it is apparent from the motion track of the fish farm that the heave motion and surge motion exhibit some attenuations with increasing draught; this is consistent with the gradual attenuation of the mooring line tension on the upstream side.

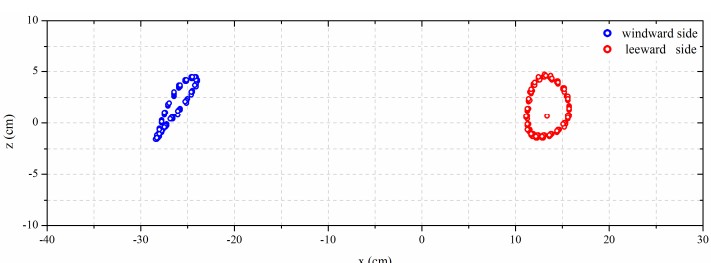

(**a**) $H = 10$ cm and $T = 1.4$ s in a draught of 28 cm.

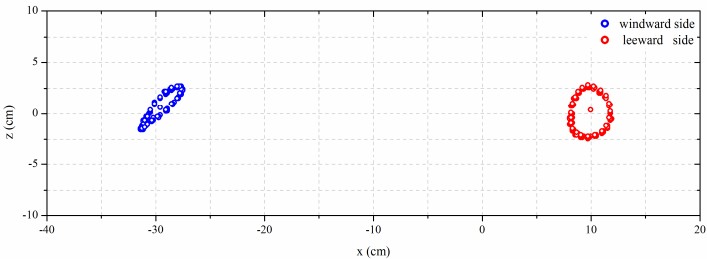

(**b**) $H = 10$ cm and $T = 1.4$ s in a draught of 36 cm.

**Figure 13.** Motion trajectories of the fish farm.

Figure 14 shows the motion response of the fish farm in different wave periods. It can be observed that the motion response is positively correlated with the wave period in different draughts. However, when the draught is 7 cm, there is no obvious correlation between the surge or pitch and the wave period. In addition, it is apparent that as the draught increases, the motion response including the heave, surge, and pitch motion turn to decrease. It can be concluded that the maximum motion response can be attained in the draught of 7 cm, followed by those in the draughts of 28 cm and 36 cm. Meanwhile, it is obvious from Figure 15 that the heave, surge, and pitch motions are proportional to the wave height and reaches their peak values at the wave height of 10 cm when the wave period is 1 s.

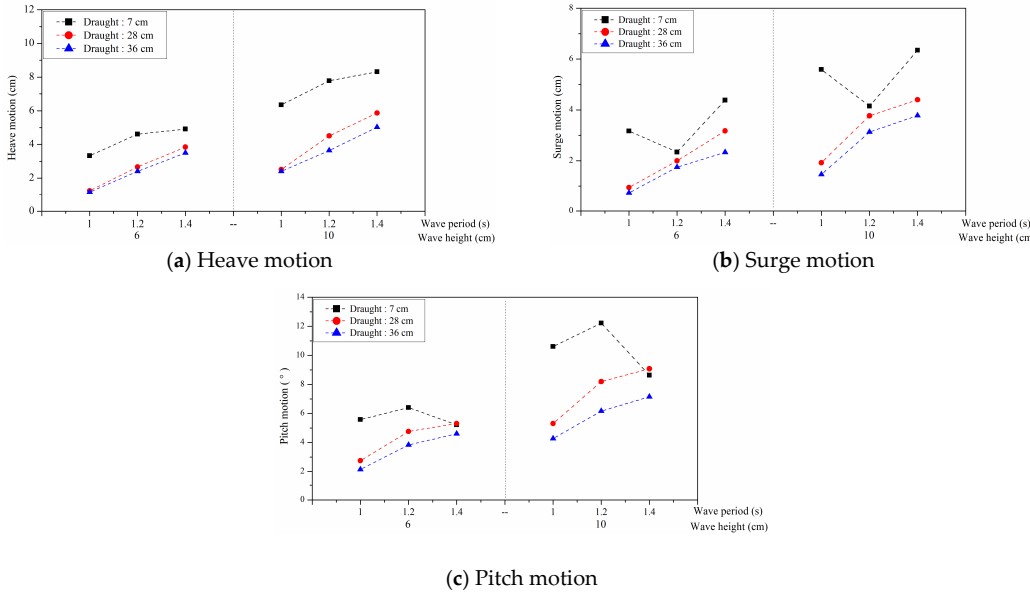

(**a**) Heave motion  (**b**) Surge motion

(**c**) Pitch motion

**Figure 14.** Motion response in different wave periods.

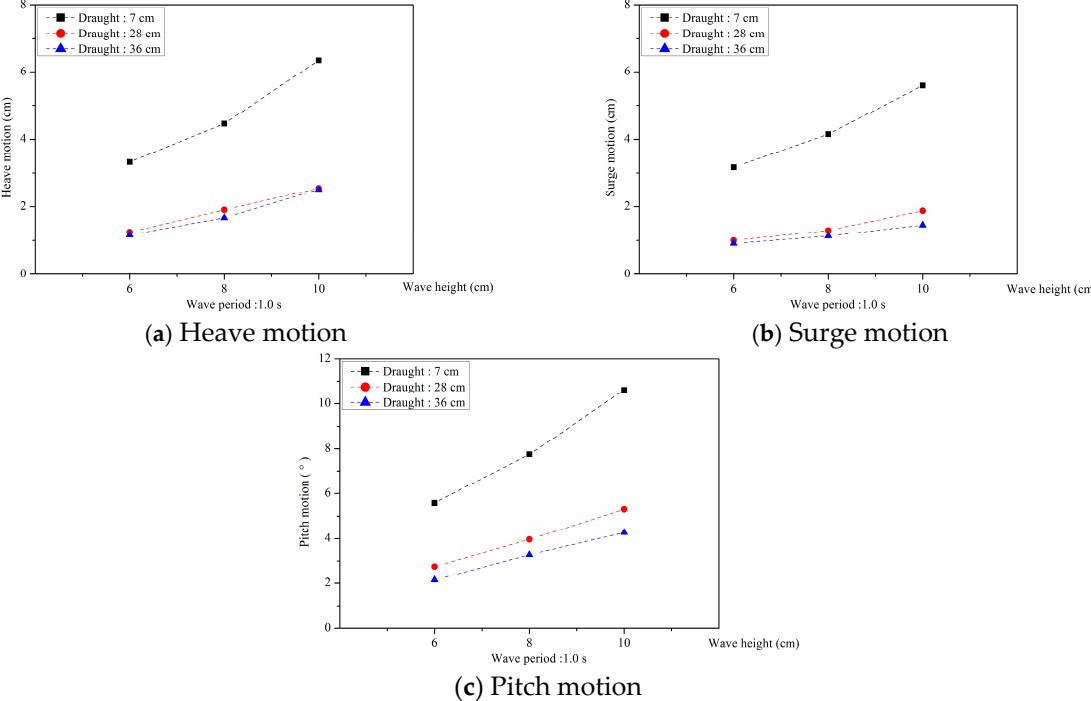

(**a**) Heave motion  (**b**) Surge motion

(**c**) Pitch motion

**Figure 15.** Motion response in different wave heights.

## 4. Discussion

The semi-submersible offshore fish farm is a new type of aquaculture facilities unlike the traditional net cage. The new fish farm has the advantages including the resistance to severe waves, small deformation, and convenient towing. To analyze the hydrodynamic response of the fish farm in detail, the wave parameters, draughts, and net damping should be considered in the research.

### 4.1. Effects of Wave Parameters

In the open sea, the fish farm is easily subjected to the wave action; therefore, the wave heights and wave periods are two important factors affecting the hydrodynamic response. As the wave heights increase, the mooring line tension and motion response exhibit an upward trend. This is because the area that can be loaded by waves gradually turn to increase with the wave heights increasing so that the motion amplitude will become larger and the mooring line will be strained, which will produce a larger mooring line tension. In addition, the results above indicate that when the draught is 7 cm, no obvious correlation exists among the mooring line tension, motion response, and wave periods. This is due to the complex pontoon that can be submerged. The surface area of the pontoon is not linearly proportional to the height along the vertical of the water surface; further, a gradual change exists in the connection between the pontoon and the column.

Taking the draught of 28 cm as an example, when the period changes from 1.0 s to 1.2 s and 1.4 s, the increment of windward force on the mooring line is from 80% to 7.6% in the wave height of 6 cm and the increment of leeward force on the mooring line is from 141% to 49% in the wave height of 6 cm. It is apparent that the trend of growth is gradually becoming slower with the increasing of wave periods. Meanwhile, the windward and leeward peak values can reach 3.37 N and 1.71 N, respectively, whose prototype values are 5823.36 kN and 2021.76 kN, respectively. In addition, when the wave height is 10 cm, the fluctuation range of the heave value ranges from 191.13% to 30.17% at $T = 1.0$ s–1.4 s; the fluctuation range of the surge value ranges from 96.15% to 16.80% at $T = 1.0$ s–1.4 s; the fluctuation range of the pitch value ranges from 54.48% to 10.79% at $T = 1.0$ s–1.4 s. It can be concluded that the increase rate of the motion response decelerates with increasing wave period. However, the whole effect on the fluctuation value of the fish farm became more evident with wave period increasing; further, when the period increases to a certain extent the surge value of the fish farm tends to be stable.

The comparison of the windward and leeward forces of the mooring line tension can be represented in Figure 16, where the wave height and wave period are 6.0 cm and 1.4 s, respectively. It can be concluded that the mooring line tension in the line 2 is less than that in line 1. Using the draught of 28 cm as an example, it can be calculated that the maximum force is attenuated by 61.7% and 70.8% from the windward side to the leeward side at $H = 6$ cm and $H = 10$ cm, respectively. It can be explained that the wave-ward lines are more loaded due to the non-linearity and drift forces when the wave reaches the fish farm. On the contrary, less external force exists on the leeward mooring line than that in the windward. Therefore, the wave loads on the fish farm acts primarily on the windward mooring line and the leeward force is relatively smaller.

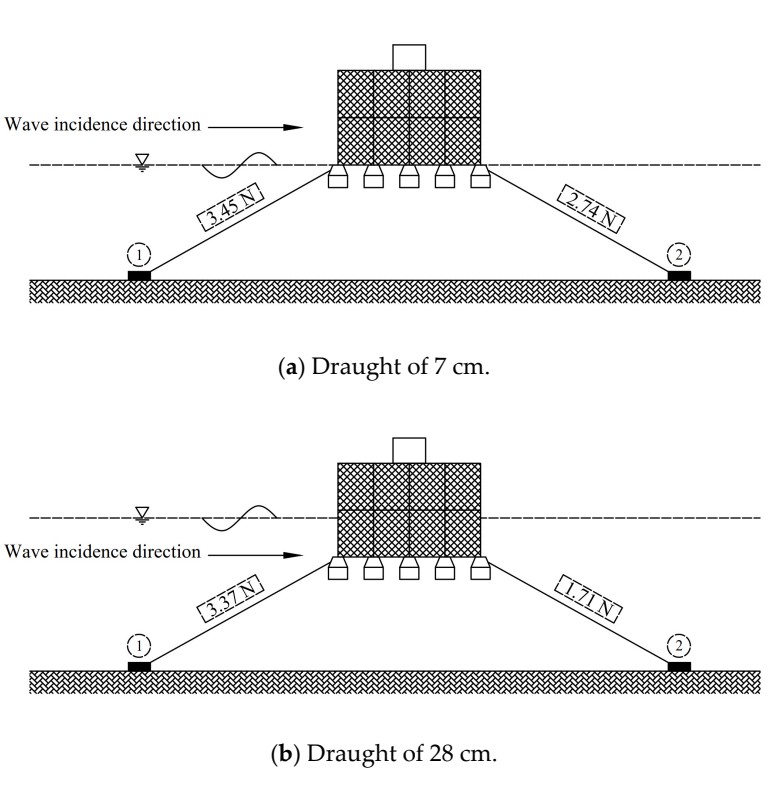

(**a**) Draught of 7 cm.

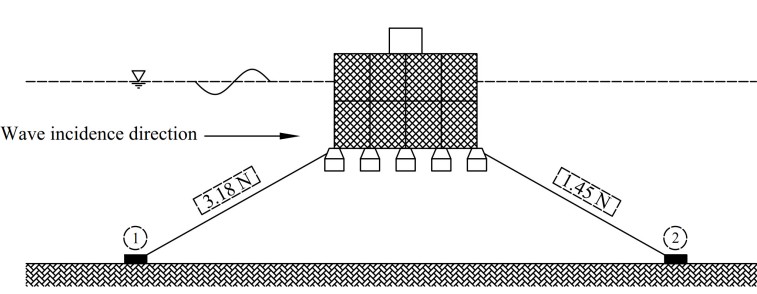

(**b**) Draught of 28 cm.

(**c**) Draught of 36 cm.

**Figure 16.** Mooring line tensions of the fish farm.

## 4.2. Effects of Draught

To overhaul and run the fish farm, different draughts can be adjusted by the weight system. With the change in draught, the force on the mooring line and the motion response will present different variation tendencies. It can be observed from Figure 17 that as the draught continues to increase, the windward force on the mooring line decreases. This is a desirable phenomenon that will provide a technical method for the security of the fish farm in severe waves. In addition, it can be calculated that the maximum reduction reaches 49.8% when the wave height is 10 cm. Meanwhile, it is apparent from Figure 18 that the motion amplitude exhibits a downward trend with increasing draught, and that the maximum attenuation including the heave, surge, and pitch motion can reach 75.6%, 73.9%, and 59.8%, respectively, when the wave height is 10 cm. This phenomenon can arise because the sloping bottom and the pontoons around the bottom contribute to the vast majority of volume of the fish farm. Therefore, a considerable wave force is produced by the interaction between the waves and structures. With the draught increases, the primary volume of the fish farm is submerged to a certain water depth and interacts with the lower velocity of the water particles. Meanwhile, the primary parts that interact with the waves are the columns in the circumference and the net chamber around the free surface that experience significantly weaker external forces. Thus, the wave force acting on the fish farm decreases

gradually. In practical implication, the draught of the fish farm can be increased appropriately to maintain the stability of the fish farm and to avoid severe wave loads from acting on the fish farm.

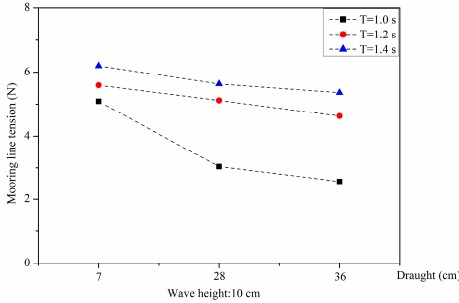

**Figure 17.** Windward mooring line tension in different draught.

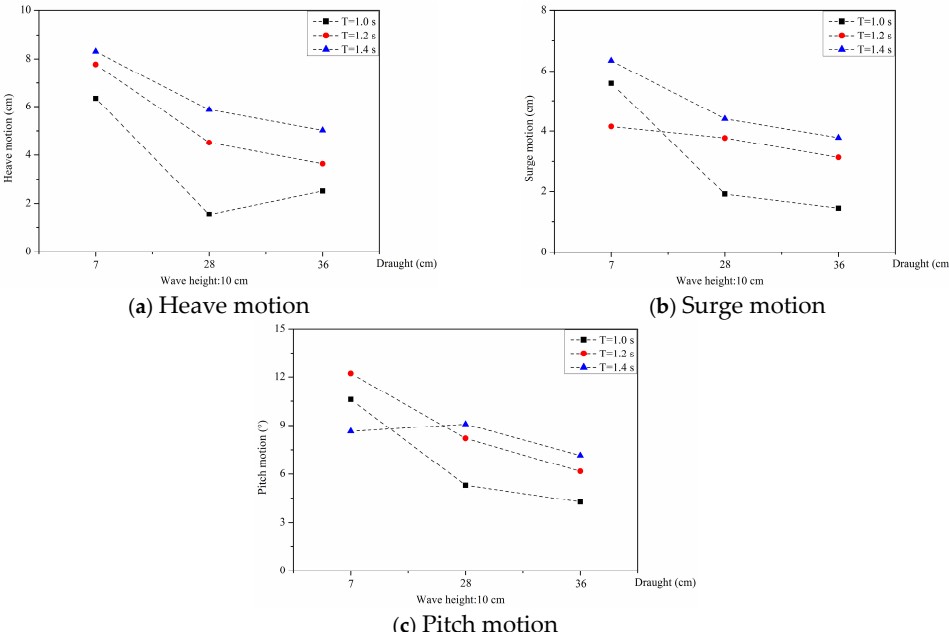

(**a**) Heave motion　　　　(**b**) Surge motion

(**c**) Pitch motion

**Figure 18.** Motion response in different draughts.

### 4.3. Effects of Net Damping

From comparing the new semi-submersible offshore fish farm and a traditional semi-submersible offshore platform, it is apparent that the primary difference is whether a net exists. The existence of net will pose certain effects on the mooring line tension and motion response of the fish farm. Figure 19 shows the comparison of the windward mooring line tension with or without net in the draught of 36 cm. It can be observed that the mooring line tension in different regular wave decreases owing to the existence of nets, and the maximum reduction can reach 42%. The phenomenon can be explained by the shielding effect and damping effect of the net [22,23]. When the wave passes the net, the wave energy will be attenuated, and the wave height reduces significantly, such that the wave loads on the leeward fish farm exhibit a downward trend. Meanwhile, the fish farm will produce a relative motion along the direction of the wave propagation so that the fluid will exhibit a greater damp effect on the net and produce a reverse force. Compared with the no-net state, the horizontal motion amplitude will be restrained to a certain extent (see Figure 20) and the mooring line tension decreases. Overall, it can be concluded that the mooring line tension will exhibit a downward trend for the fish farm with a net compared with no net.

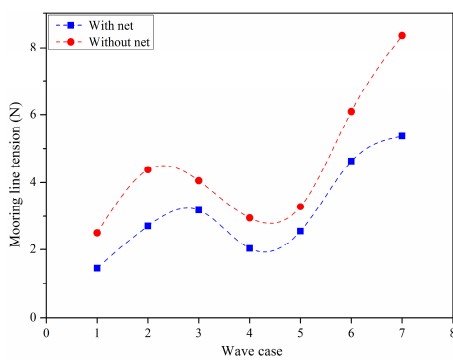

**Figure 19.** Comparison of mooring line tension.

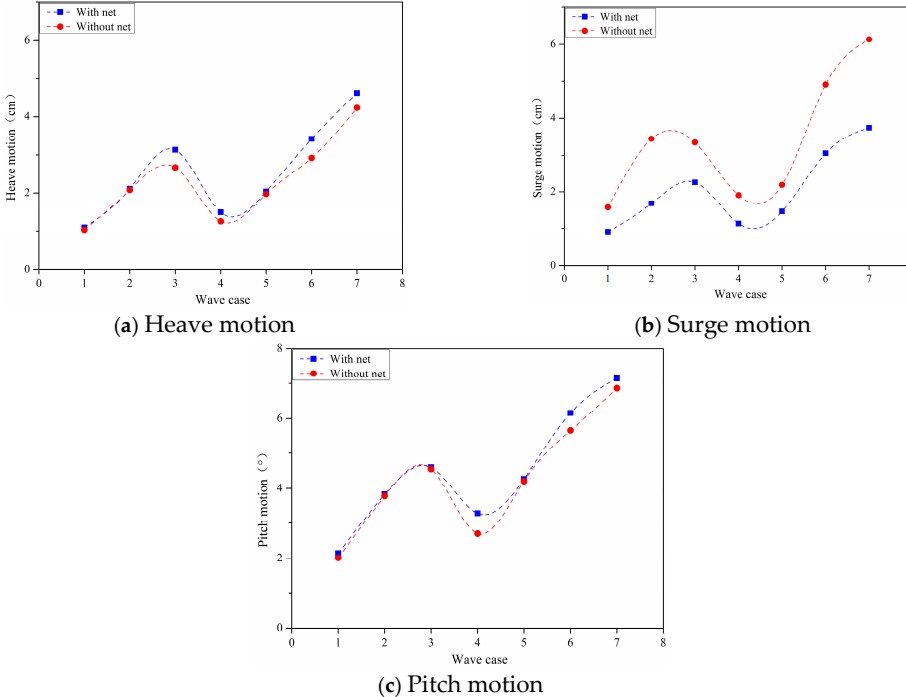

**Figure 20.** Comparison of motion response.

As for different wave conditions, the change trend of surge motion of the fish farm is the same as that of the mooring line tension. The surge motion will reduce for the fish farm without net. This is because the fluid damping will produce an important impact on the net. Overall, the surge motion of the fish farm with net is less than that without net, and the maximum difference can reach 51%. In addition, the heave motion and pitch motion of the fish farm with net are larger than that without net, and the maximum difference can reach 19% and 22%, respectively. It can be concluded from Figure 20 that the existence of net affects the pitch motion less but affects the surge motion more.

In the paper, the hydrodynamic response was studied and analyzed in the pure waves. Due to the existence of waves, the motion response of the fish farm will be significantly affected. However, in the actual sea conditions, waves and currents always exist at the same time and have an interaction with each other. The existence of water flow is equivalent to adding an external load on the fish farm, which will have a greater impact on the anchor rope force. Therefore, in the next work, the hydrodynamic characteristics of the fish farm under the combined action of waves and currents will be considered, which will provide suitable suggestions for the application of practical projects.

## 5. Conclusions

In the present study, the hydrodynamic responses of a semi-submersible offshore fish farm in regular waves were investigated at various draughts using physical model experiments. The mooring line tension and the motion response of the fish farm were analyzed. The conclusions are summarized as follows:

(1) The mooring line tension and motion response were closely proportional to the wave period and wave height. Nevertheless, when the draught was 7 cm, no obvious correlation existed between the hydrodynamic response and wave periods. In addition, the force on the windward mooring line was higher than that on the leeward; further, the upstream anchor lines endured most of the external loads on the fish farm. Therefore, the upstream anchor lines should be optimized and materials with better mechanical properties should be chosen in the actual operation.

(2) As the draught increased, the mooring line tension and motion response exhibited a slight downward trend. Although the area of interaction between the fluid and structure increased, the range of wave action was near the sea surface; subsequently, the effect became weaker gradually with increased draught. In addition, owing to the damping effect of fluid, the motion of the fish farm was limited and the force on the mooring line exhibited a slight downward trend. The experimental results provided a good method to avoid severe waves.

(3) The existence of net affected the hydrodynamic characteristics of the fish farm significantly. Compared with the fish farm without net, the mooring line tension exhibited an obvious increase and the reduction in the peak value could reach 42%. Because of the shielding effect and damping effect of the net, the surge motion of the fish farm exhibited a maximum attenuation of approximately 51%. However, the heave motion and pitch motion increased to some extent. It can be concluded that the net affected the surge degree the most but affected the heave and pitch degrees the least.

(4) Pure waves have a great influence on the motion response of fish farm and is a critical factor affecting the stability of fish farm, so it is important to design a suitable parameter to decrease or avoid the large motion of the fish farm. However, in actual sea conditions, the presence of water flow will increase the anchor rope force and affect the waves. Therefore, in the following work, the hydrodynamic response of the fish farm under the waves and currents will be studied to provide some experimental guidance for the practical engineering.

**Author Contributions:** Data curation, C.B., H.L. and Y.C.; formal analysis, C.G., H.L. and Y.C.; funding acquisition, Y.Z. and C.G.; investigation, Y.C.; methodology, C.B.; resources, C.G.; software, Y.C.; supervision, Y.Z.; writing—original draft, Y.Z., C.B., and H.L.; writing—review and editing, Y.Z., C.G., C.B., and H.L.

**Funding:** This research was funded by the National Natural Science Foundation of China (NSFC), grant number 31872610, 51822901, 51609035, 51579037 and 31772898; and Fundamental Research Funds for the Central Universities, project nos. DUT19ZD206, and DUT18RC (3) 076.

**Conflicts of Interest:** The authors declare no conflict of interest.

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
