# Peer review of "Experimental Investigations on Hydrodynamic Responses of a Semi-Submersible Offshore Fish Farm in Waves"

_jmse, doi:10.3390/jmse7070238_

Round 1
Reviewer 1 Report
1. The value of this paper is mostly due to the applicability of results for designers of fish farms structures and mooring. Hence, the prototype and model structure and mooring need to be completely specified. I found the following data missing:
1.1. Prototype structure material (I guess steel, however what kind specifically).
1.2. In table 1, I did not find the central column and the upper Brace.
1.3. In table 1, the wall thicknesses for all components need to be specified.
1.4. In table 1, it is not likely that the length of the inclined column equal to the vertical.
1.5. In table 1, the term "Table length" of the pontoon is not clear to me.
1.6. I suggest adding an engineering drawing of the prototype.
1.7. Please show more specifically the locations of connections of mooring lines.
1.8. Specification of mooring system of the prototype: mooring lines material, diameter, breaking load; anchors.
1.9. The design environmental conditions of the prototype.
2. Scale effects
The model similarity is based on Froude number, which is proper for inertia loads and high Reynolds numbers (full turbulence flow) drag loads. However, the structure contains thin pipes and nets of very thin yarn, for which considerable scale effects are expected. Please discuss this issue and justify that the scale effects are acceptable for design. How do you consider the scale effects in the interpretation of model results to prototype, and how?
3. Current
The conditions of wave with no current are, to my opinion, not practical for design. Current loads to the net are expected to be critical. Although there are simple methods to calculate the current loads, they cannot be added to the wave loads from the experiments, as the current will tense the mooring system and affect the wave loads. As the laboratory experiments were performed in a wave–current flume, I think that it is critical to perform wave–current experiments.
4. Pre tension
The pre tension of the mooring lines (at calm sea with no waves and currents) at each draft should be specified.
5. What is the sampling rate of mooring line tension and of displacements?
6. I suggest adding typical figures of the time series for the mooring lines tensions and all the displacements simultaneously. The line tension is determined by the position of its connection to the structure, and such relations may be analyzed and discussed for checking the accuracy of results and understanding the mooring design. Such a data processing can improve the study.
7. Please show on the drawing, with dimensions, the location of tracing points for motion responses.
8. To my opinion, the discussion in Item 4.1, lines 356-362 is not correct. The waves move the whole rigid structure and the locations of the connections of the mooring determine the tensions in the mooring lines by their load-elongation curves. The tension in a mooring line is not related to the local attenuation of waves in its vicinity. The wave-ward lines are more loaded due to the nonlinearity and drift forces (in ideally linear solution for regular waves the amplitudes of motion will be the same for each side and the maximum tension will be equal in all lines).
Author Response
Dear Reviewer:
Thanks for your suggestion. The resoponse has been writing in the PDF files. Please see the attachment

Reviewer 2 Report
The manuscript contributes to the development of offshore aquaculture.
There are a few corrections and suggestions.
Line 30 – The authors wrote “offshore aquaculture is becoming prevalent”. Is it prevalent or is it growing?
Line 43 – The Authors wrote “offshore fish farm is bound to be accepted and widely used”. At the moment there are studies, like the one that is presented in this manuscript, that are conducted to test and to investigate the structures. It means that at the near future, most probably, they can not be used all over the world. The Authors should not be so sure about what, or when, it will happen. Also, more studies about ecology and epidemiology are needed. The study of diseases and fish escapees are important and can limit offshore aquaculture.
Line 107 – Why is it written “and so on”?
Lines 124-125 – The sentence “the elastic similarities need not be considered.” It should be explained.
Figure 4 – In the Figure 4, it is written “Adding sands”, Injecting water” and “Hanging weights”. I suggest that it can be replaced by numbers (1, 2 and 3) and in the legend the Authors can identify and explain them using correct language sentences.
Why is it written “Adding sands”? Why is the word “sand” written in the plural?
Line 192 – Why is it written “It is obvious that the error is small”? Is it “obvious” or is it expected?
Lines 284-285 – The Authors wrote “the motion response is critical to the fish living in the fish farm and human activities”. Why is it critical for the fish?
Line 287 – It is written “three aspects the heave, surge, and pitch.” The sentence should be corrected, namely including “:” after “aspects”.
Lines 323-324 – The Authors wrote “The new fish farm combines the advantages of the traditional semi-submersible fish farm”. The sentence suggests that the “traditional” have the same advantages of the “new”.
Line 397 – The Authors wrote “comparing the fish farm and a traditional fish farm”. The “traditional fish farm”, is it inland water cage or pen fish culture? Is it offshore or inland aquaculture? Does it refer to floating cages, to submersible cages, to semi-submersible cages or to earth ponds?
The “fish farm” should be clear in the text.
Lines 409-410 – In the manuscript, it is written more than one time (also in Line 419) “fish farm with and without net”. I consider that the structure can have or not a net. If the structure contains fish, a net is mandatory otherwise it is not a fish farm or a fish cage.
Author Response
Dear Reviewer:
Thanks for your suggestion. The resoponse has been writing in the PDF files. Please see the attachment.

Round 2
Reviewer 1 Report
1. The value of this paper is mostly due to the applicability of results for designers of fish farms structures and mooring. Hence, the prototype and model structure and mooring need to be completely specified. I found the following data missing:
1.1. In table 1, the wall thicknesses for all components need to be specified for the Prototype (those of the model are not of much interest).
1.2. Specification of mooring system of the prototype is mandatory. The model cannot represent the prototype if the response curves (load-elongation) and the geometry of the mooring are not properly scaled.
2. Scale effects
As the reference is in Chinese, please extend the explanation in this paper.
3. Current
Please discuss the practical value of experiment with waves only, and indicate the future research.
Author Response
Dear reviewer:
Thanks for your suggestion. The paper has been revised and please see the attachment.

Round 3
Reviewer 1 Report
1. In table 1, I guess the wall thicknesses for the Prototype are in mm, while m is indicated for the whole column.
Author Response
Dear Editor -in -ChiefChief ,
Thanks for your suggestion, The mistakes have been revised in the manuscript. Please see the attachment.